# *Anopheles gambiae* Genome Conservation as a Resource for Rational Gene Drive Target Site Selection

**DOI:** 10.3390/insects12020097

**Published:** 2021-01-23

**Authors:** Nace Kranjc, Andrea Crisanti, Tony Nolan, Federica Bernardini

**Affiliations:** 1Department of Life Sciences, Imperial College, London SW7 2AZ, UK; n.kranjc@imperial.ac.uk (N.K.); a.drcrisanti@imperial.ac.uk (A.C.); 2Department of Vector Biology, Liverpool School of Tropical Medicine, Liverpool L3 5QA, UK

**Keywords:** *Anopheles gambiae* complex, vector control strategies, gene drive, genetic variability, genome conservation, genetic resistance

## Abstract

**Simple Summary:**

Malaria is a huge public health burden that affects predominantly sub-Saharan Africa and is transmitted by *Anopheles* mosquitoes. As a measure for population control, a method called gene drive has been recently developed, which relies on genetic engineering to introduce specific genetic traits into mosquito populations. Gene drives are designed to insert at specific target sites in the mosquito genome. The efficacy of gene drives greatly depends on the selection of appropriate target sites that are functionally or structurally constrained and less likely to tolerate mutations that can hinder the spread of the desired trait in the population. The aim of this study was to perform a genome-wide analysis of highly conserved genomic regions in *Anopheles gambiae* and introduce a measure of conservation that could indicate sites of functional or structural constraint. The results of this analysis are gathered in a publicly available dataset that can support gene drive target selection and can offer further insights in the nature of conserved genomic regions.

**Abstract:**

The increase in molecular tools for the genetic engineering of insect pests and disease vectors, such as *Anopheles* mosquitoes that transmit malaria, has led to an unprecedented investigation of the genomic landscape of these organisms. The understanding of genome variability in wild mosquito populations is of primary importance for vector control strategies. This is particularly the case for gene drive systems, which look to introduce genetic traits into a population by targeting specific genomic regions. Gene drive targets with functional or structural constraints are highly desirable as they are less likely to tolerate mutations that prevent targeting by the gene drive and consequent failure of the technology. In this study we describe a bioinformatic pipeline that allows the analysis of whole genome data for the identification of highly conserved regions that can point at potential functional or structural constraints. The analysis was conducted across the genomes of 22 insect species separated by more than hundred million years of evolution and includes the observed genomic variation within field caught samples of *Anopheles gambiae* and *Anopheles coluzzii*, the two most dominant malaria vectors. This study offers insight into the level of conservation at a genome-wide scale as well as at per base-pair resolution. The results of this analysis are gathered in a data storage system that allows for flexible extraction and bioinformatic manipulation. Furthermore, it represents a valuable resource that could provide insight into population structure and dynamics of the species in the complex and benefit the development and implementation of genetic strategies to tackle malaria.

## 1. Introduction

Malaria is a threat to life, health and wellbeing. It is a parasitic disease transmitted exclusively by mosquito species from the *Anopheles* genus. In 2018 there were 228 million cases recorded worldwide and 405,000 of these resulted in death [1]. The investment in malaria control programs relying on insecticide-treated mosquito nets (ITNs), indoor residual spraying and artemisinin-based combination drug therapies has been crucial for the decline of malaria incidents between 2000 and 2015, with the vast majority of this decline being attributed to the control of mosquito numbers [2]. However, from 2014 to 2018, the rate of change slowed dramatically due to the rise of resistance to drugs and insecticides [1].

In sub-Saharan Africa, where the malaria burden is highest, the majority of malaria transmission is caused by members of the *Anopheles gambiae* species complex. Eight species are currently included in this complex of which three, *Anopheles gambiae*, *Anopheles coluzzii* and *Anopheles arabiensis*, are responsible for 95% of malaria transmission in Africa [3]. Recently, a new species, *Anopheles fontenillei*, was discovered in Central Africa and attributed to the complex [4]. Transgenesis tools for *Anopheles* mosquitoes have been in use since the turn of the century [5,6] and in recent years, a number of genetic control methods have been generated that have proven, in a laboratory setting, to have great potential to tackle malaria [7,8,9,10,11]. Among the genetic control methods available for suppression or alteration of *Anopheles* mosquito populations, technologies based on gene drive systems seem to be the most promising. Naturally occurring gene drives alter Mendelian rules of segregation by increasing the probability that a specific allele will be included in the gametes that contribute to the offspring of any one individual. This higher than normal rate of inheritance allows the rapid spread of that specific allele through the population [12,13,14].

In recent years, genome editing technologies have facilitated the development of synthetic gene drives designed to enable the spread of altered traits mimicking the dynamics of the gene drives found in nature [10,11,15,16,17]. Due to their simple molecular components and efficiency, CRISPR/Cas9-based homing gene drives have quickly become a system of choice in this context. This technology is based on the presence of a bacterial endonuclease (Cas9) and a guide RNA (gRNA) that is responsible for directing the sequence specificity of Cas9 [18]. A genetic construct encoding the units of the Cas9-gRNA complex can be inserted into the genome of an organism to cause disruption of a target gene. The driving nature of this system relies on the ability of the gRNA to direct Cas9 to cut the wild-type version of the target gene on the homologous chromosome. This cut triggers the activation of the cell DNA break repair machinery, which will use the homologous regions flanking the genetic construct as a template to repair the double strand break. The repair mechanism can thus lead to the copy of the gene drive construct (and any encoded trait) and the conversion of the cell to a homozygous genotype. When this process happens in the germline it results in gametes with a higher than 50% probability of containing the gene drive, which in turn causes the allele to increase in frequency each generation and spread through the population. We have previously shown that a CRISPR-Cas9 gene drive construct can be used to target genes with essential roles in the female fertility of *Anopheles* mosquitoes [9]. However, the high fitness cost imposed by this type of gene drive can create a strong selection pressure for mutations that restore function to the target gene yet are resistant to CRISPR-Cas9 cleavage. This highlights the requirement, when designing gene drive systems, for a meticulous selection of genic target sequences that are under high levels of functional constraint and therefore less likely to tolerate resistant mutations. Indeed, in caged populations, suppression was achieved only when *doublesex*, a gene involved in sex determination and highly conserved within the *Anopheles* complex was targeted, showing that this aspect is crucial for the success of mosquito population control through this approach [11,19,20,21].

High levels of sequence conservation can be a good proxy for functional constraint. Recently, high-throughput DNA sequencing of wild-caught *Anopheles* mosquitoes revealed complex population structure, patterns of gene flow and a genetic diversity higher than expected [22,23,24]. Here we describe a bioinformatics pipeline that integrates a systematic analysis of the data on genetic variation in more than 1000 wild-caught *Anopheles gambiae* individuals and conserved syntenic regions of 19 *Anopheles* species, and three phylogenetically more distant species within the same Diptera order. This user-friendly pipeline performs a sliding window analysis across the whole genome for every possible gene drive target sequence, attributing to it a conservation score (Cs), which is an indicator of their tendency to tolerate mutations. Cs incorporates interspecies variation, intraspecies variation and a measure of the selection forces applying there. This should prove a useful resource when selecting for functionally constrained target site within genes of interest in the context of vector control via gene drive. In addition, the compiled dataset offers the flexibility to identify conserved genomic regions within species subsets, which will be of significant interest for the implementation of any vector control strategies in geographical areas populated by sympatric species and where one wishes to target only a particular species.

## 2. Materials and Methods

### 2.1. Genomic Data

Genomic data of 19 *Anopheles* species, *Aedes aegypti*, *Culex quinquefasciatus* and *Drosophila melanogaster* were used in the analysis. The following reference genomes were obtained from VectorBase [25]: AgamP4 (*An. gambiae*), AcolM1 (*An. coluzzii*), AaraD1 (*An. arabiensis*), AquaS1 (*An. quadriannulatus*), AmelC2 (*An. melas*), AmerM2 (*An. merus*), AepiE1 (*An. epiroticus*), AchrA1 (*An. christyi*), AsinC2 (*An. sinensis*), AminM1 (*An. minimus*), AmacM1 (*An. maculatus*), AculA1 (*An. culicifacies*), AsteI2 (*An. stephsnsi*), AfunF1 (*An. funestus*), AatrE3 (*An. atroparvus*), AdirW1 (*An. dirus*), AfarF2 (*An. farauti*), AdarC3 (*An. darlingi*), AalbS2 (*An. albimanus*), AaegL5 (*Aedes aegypti*), CpipJ2 (*Culex quinquefasciatus*). The reference genome of *Drosophila melanogaster* DmelP6 was obtained from Flybase [26].

### 2.2. Genome Alignment

To identify conserved regions, we used the CNEr package [26], which can be used to detect highly conserved noncoding elements (CNE). CNEr streamlines the process of large-scale identification of CNEs. CNEr uses LASTZ genome aligner together with *axtChain*, *chainMergeSort*, *chainPreNet*, *chainNet*, *netSyntenic*, *netToAxt* and *axtSort* tools for chaining and netting. Genome alignment, chaining and netting were performed by using default parameters (C = 0 E = 30 H = 0 K = 3000 L = 3000 M = 50 O = 400 T = 1 Y = 9400). All of the selected genomes were aligned and scanned for highly identical regions relative to *An. gambiae* AgamP4 genome with the pipeline provided by CNEr. Two sets of identity thresholds were used: 90%, 97%, 100% and 70%, 90%, 96%, 98%, 100% identity over a scanning window of 30 and 50 bp, respectively. Although CNEr was designed for detection of conserved noncoding elements, by omitting the filtering step it retained detected conserved elements in the coding regions as well. This procedure yielded a list of conserved genomic regions between each selected genome and *An. gambiae* where the sequence identity was higher than a threshold set in each scan. The minimal detected identity was set to 70%. For example, an identity value of 98% corresponds to detecting 49 identical positions in the 50 bp alignment window.

### 2.3. Conservation Score Calculation

Genomic intervals with their corresponding CNEr identity values were mapped to a matrix of the size of each chromosome, allowing for per-base-pair resolution of identity scores and easier comparison across different species. Additionally, positions of SNPs found in 1142 wild-caught individuals within the Ag1000g phase 2 project [23,27] were mapped in the same data table and average SNP density in the sliding window of 20 bp was calculated across all chromosomes.

To calculate an overall conservation score (Cs), identity values were normalised with the phylogenetic distance between each species and *An. gambiae.* The phylogenetic tree was generated from the genome alignments using PHAST phyloFit [28]. Phylogenetic distances within the tree were calculated using ETE Toolkit [29]. Average normalised identity values were calculated for each position and SNP density data were used to skew values higher when SNP density was low. Furthermore, phyloP was used to calculate scores with likelihood ratio test (method LRT) representing conservation or acceleration (mode CONACC) for each genomic position. Incorporating phyloP scores into the model indicated whether positions within highly identical genomic regions are under negative or positive selection, based on the phylogenetic model of neutral evolution (REV substitution model).

A matrix of normalised identities was created (Equation (1)), where index m is the number of genomes in comparison, index n is the length of a chromosome and $d_{m}$ is the phylogenetic distance between genome m and AgamP4.
(1)$${\tilde{I}}_{m,n}=\left[\begin{array}{ccc} I_{1,1} & \cdots & I_{1,n}\\ \vdots & \ddots & \vdots \\ I_{m,1} & \cdots & I_{m,n} \end{array}\right]\left[\begin{array}{c} d_{1}\\ \vdots \\ d_{m} \end{array}\right]$$

Normalised identities were averaged for each nucleotide position n in each chromosome. SNP density σ calculated for a sliding window of 20 bp on each position n. PhyloP score for position n is denoted by $p_{n}$ (Equation (2)).
(2)$$Cs_{n}=\frac{\sum_{i=0}^{m}{\tilde{I}}_{i,m}}{m}\;\frac{1-\;\sigma_{n}}{1+\;\sigma_{n}}\;\frac{1}{2^{-p_{n}}}$$

Finally, *Cs* values were scaled using MinMax scaler to a range between 0 and 1, where 1 means high conservation per chromosome arm. Normalisation was also performed on the whole genome and showed negligible difference in values. All calculations were done using Python modules Pandas, Numpy, Scipy, Zarr, h5py, scikit-learn and scikit-allel [30]. The data were stored in HDF5 format and plotted using Matplotlib and Seaborn. Genome accessibility data were obtained from Ag1000g phase 2 [23,27].

The dataset produced in the analysis can be found at https://zenodo.org/record/4304586. A Python tool for easier accessibility of the results was created and can be found together with the scripts used to generate the dataset at https://github.com/nkran/AgamP4_conservation_score.

To account for the missing data, a list of regions and the multisequence alignment depth, indicating the number of analyzed assemblies aligned across the genome, was gathered and deposited at https://github.com/nkran/AgamP4_conservation_score/tree/master/data.

## 3. Results

### 3.1. Conservation Score Calculation

We selected available reference genomes of 18 *Anopheles* species, namely *An. coluzzii, An. arabiensis, An. quadriannulatus, An. melas, An. merus, An. epiroticus, An. christyi, An. sinensis, An. minimus, An. maculatus, An. culicifacies, An. stephensi, An. funestus, An. atroparvus, An. dirus, An. farauti, An. darlingi, An. albimanus*, as well as reference genomes of the more distantly related species from the order Diptera: *Aedes aegypti, Culex quinquefasciatus* and *Drosophila melanogaster*. Genome sequences of each species were aligned to *An. gambiae* reference genome to find syntenic regions for each pair. Syntenic regions obtained from the alignments were analyzed to identify genomic intervals of highly conserved regions (>70%) relative to *An. gambiae*. We then calculated the average identity across all species included in the analysis and normalised the values based on the phylogenetic distance between *An. gambiae* and the other species. Normalisation was performed to balance the sequence identity scores, giving more weight to the same identity scores in species that are phylogenetically more distant than the species that are phylogenetically closer to *An. gambiae*. 

To account for the evolutionary selection, we incorporated a measure of positive or negative selection, PhyloP score, in the calculation of Cs for each position in the genome. The PhyloP score is calculated by taking into account a substitution model of neutral selection and it is used to perform a statistical test at each position in the multiple alignment for a significant increase or decrease in the rate of substitution.

Additionally, intraspecies variation was included in the calculation of the final conservation score (Cs), where a value of Cs = 1.0 represents loci with the highest conservation. Cs was generated as a measure of conservation with a base-pair resolution that takes into account genomic variation of compared species and intraspecific variation observed in 1142 wild-caught mosquito individuals.

To investigate differences in genome conservation of different species, we directed our analysis toward two groups, one including five sibling species within the *Anopheles gambiae* complex (*An. coluzzii, An. arabiensis, An. quadriannulatus, An. melas, An. merus*), and the other including five more distantly related species (*An. darlingi, An. albimanus, Aedes aegypti, Culex quinquefasciatus, Drosophila melanogaster).*

The genome identity at the chromosomal level was investigated and compared between the two clusters. We calculated mean identity of 81.0% across the genomes of species within the *Anopheles gambiae* complex (84.4% in autosomes and 67.2% in X chromosome) and 4.8% for the genomes of distantly related species (5.12% in autosomes and 3.69% in X chromosomes). Both groups showed decreased conservation toward the centromeric end of each chromosome and on X chromosome compared to autosomes (Figure 1).

Additionally, to test the bias introduced by the inclusion of the five more distantly related species on the overall Cs performance, Cs was calculated by omitting these species from the analysis. A strong correlation (Pearson’s *r* > 0.83, *p* < 0.001) was observed between both Cs calculations. Inclusion of the five species skews the Cs values, however the highest-ranking positions are not considerably affected (Figure A2).

### 3.2. Conservation of Gene Drive Target Sites

Homing-based gene drives targeting female fertility have shown varying degrees of efficacy, in part related to the propensity of the target genes to tolerate resistant mutations. With the rationale that those target genes may have additional gene drive target sites less prone to the development of resistance, and as a proof of principle for the pipeline developed herein, we scanned for alternative target sites and compared conservation scores among them in order to identify possible regions of high functional constraint. We focused our analysis on two genes previously selected for the development of gene drive technologies against *An. gambiae* mosquitoes; the female-fertility gene *AGAP007280*, whose original target site was very prone to resistance, and *doublesex*, a gene whose female-specific isoform can be targeted by a gene drive without, to date, selection for resistance.

Consistent with the observation that it could tolerate a range of viable mutations, the original 18 bp-long target site that was chosen to disrupt gene function of the female fertility gene *AGAP007280* (2L:44994011–44994029) had a low conservation score with a mean Cs = 0.06 (Cs_min_ = 0.01, Cs_max_ = 0.24) (Figure 2A). We calculated the conservation score for all 987 SpCas9 accessible target sites in this gene. We found an alternative target site in exon 6 (2L:44993662–44993685) with the highest mean conservation score of 0.44 (Cs_min_ = 0.05, Cs_max_ = 0.73). Mean Cs of 0.06 and 0.44 correspond to the 57th and 94th percentile of all Cs values (Cs > 0), respectively, for chromosome arm 2L (Figure A1). This target site is located in a locus that is part of a conserved region annotated as peptidase S1 domain, which spans exons 4, 5 and 6.

In comparison, the previously validated gene drive target site spanning intron 4 and the female-specific exon (exon 5) of the *dsx* gene (*AGAP004050*), shown to be essential for female sex determination and intolerant of mutations, showed high mean Cs of 0.43 (Cs_min_ = 0.17, Cs_max_ = 0.68). The same site was 100% conserved across the *Anopheles gambiae* complex and showed high conservation for three out of the five distantly related species (92.8%, 86.5%, 86.2% in *An. albimanus, Aedes Aegypti* and *Culex quinquefasciatus*, respectively). Interestingly, the whole of exon 5 has a relatively high mean conservation score (Cs = 0.50) across its relatively short (87 bp) length with only one SNP present (Figure 2B). This allowed us to identify eight alternative SpCas9 accessible sites in exon 5 (Table 1) of which the highest-ranking alternative target site (2R:48714594–48714616) had an even higher conservation score Cs = 0.60 (Cs_min_ = 0.26, Cs_max_ = 0.70) than the original gene drive target site. Both mean Cs values for the previously validated gene drive target site and the alternative target site in exon 5 of *dsx* are in the top 10th percentile of all Cs values for chromosome arm 2R.

The high level of sequence conservation at the gene drive target site within dsx gene that was observed across species of the *Anopheles* complex as well as more distantly related species, gave the rationale for their inclusion in the analysis.

## 4. Discussion

The availability of whole genome data for species of the *Anopheles gambiae* complex has enabled us to conduct a bioinformatic analysis where a conservation score, which accounts for interspecies and intraspecies variation, was attributed to genomic regions across these species as well as to more distantly related ones.

A strong pattern of conservation was observed for species of the *Anopheles* complex with an average genome identity of 84% across the autosomes and lower values only falling in those regions of the chromosome annotated as centromeric. This discrepancy is in agreement with a number of works that describe centromeric sequences as highly variable in size and composition [31,32]. Overall, a lower average sequence identity (67%) was observed for the X chromosome, reflecting the expectation that genetic divergence tends to be higher on sex chromosomes [33,34] and confirming the previously shown diversity of this chromosome within the species of the complex [24,35,36,37,38]. This conservation pattern across the more distantly related species, albeit at a lower level (4.8%), is consistent with longer evolutionary distances that separate them [24]. These observations confirmed the suitability of using sequence identity deriving from multiple alignment as the base of the proposed conservation model, which was further improved by including the intraspecies variation.

It has recently been shown that, despite high genetic variation between genomes of *Aedes* and *Anopheles* mosquitoes, conserved target sites for Cas9-based gene drives are abundant within protein-coding sequences [39]. We made use of existing bioinformatic methods to account for positive or purifying selection on the genomic sequences used in this study [28] and strengthened our analysis by developing a bioinformatic pipeline that relies on intraspecies genetic variation as well as highly conserved sequences of syntenic regions between several species of dipterans for the identification of ultraconserved regions. This analysis offers a flexible workflow where conserved regions can be identified on the genome-wide scale as well as with per base-pair resolution.

For some previous CRISPR-based gene drives designed to interrupt female fertility genes (e.g., *AGAP007280*), genotypic analysis of mosquito samples collected during cage trial experiments tracking the gene drive over time showed that end-joining repair of CRISPR-induced DNA breaks at the selected target site generated resistance, thus preventing the further spread of the gene drive [9,40]. Highlighting the utility of our pipeline, we were able to identify additional gene drive target sites in *AGAP007280* with a conservation score much higher than the one originally chosen (Cs 0.42 versus 0.06) (Figure 2A). Notably, this score is similar to the conservation score attributed to the previously validated target site in *dsx*, which has so far failed to yield resistant alleles, consistent with this region of the gene showing functional or structural constraint. Thus, our pipeline offers additional target sites, likely less prone to resistance, in genes previously validated as suitable for targeting by population suppression such as *AGAP007280*. The ability to multiplex gene drives by incorporating several gRNAs recognizing separate target sites in the same target gene is anticipated to further delay the onset of resistance since the probability of individuals being resistant at both target sites is much lower [41,42,43,44]. Thus, it is prudent to select multiple sites, each with the highest conservation score where possible. In this respect it is encouraging that alternative target sites with high conservation scores were found when exon 5 of *dsx* gene was analyzed (the highest-ranking target site had Cs = 0.60). The identification of these sites through our pipeline should thus facilitate the design of multiplexed gene drive constructs targeting the *dsx* gene and attempts to future-proof these constructs for release in the field—although to-date the original single target site fails to yield resistant alleles in the laboratory, the situation in the wild, at increased geographical and temporal scale, and with larger population sizes, may be more challenging.

The conservation scores described herein are a good indicator for conserved genomic regions under functional or structural constraint. That said, at any two functionally constrained sequences that show ultraconservation the actual level of constraint applying at each might be considerably different—the force of purifying selection could be sufficient to ensure conservation of sequences where changes caused minimal or intermediate reductions in fitness as it would at sequence changes that were nonviable. The choice between target sites at each extreme of functional constraint very much depends on the type of gene drive approach, and the intended effect of the drive.

It must also be noted that confounding sources might lead to ultraconservation that do not necessarily reflect functional constraint. For instance, repetitive regions of the genome and their representation in the reference genome assemblies can lead to ambiguous genome alignments and false identical regions. We used genome accessibility data generated in the Ag1000g project that includes already annotated, masked, low-complexity genomic regions and DNA sequences with low coverage and poor mapping quality [22,23] to label these regions in the analyzed genes. High conservation score in regions marked as inaccessible (Figure 2) should be considered and taken cautiously as indicator of functional constraint. In addition, many families of repetitive sequences, such as satellite DNA and ribosomal DNA genes, are maintained homogenous within populations by concerted evolution rather than unbreakable constraint [45]. Concerted evolution for such elements should be taken into account when conservation within and among different species is evaluated.

The understanding of the real constraint within the ultraconserved regions identified in this study represents our next step to satisfy the increasing request for suitable target sites for genetic vector control strategies based on gene drive as well as for other approaches to vector control. Furthermore, the dataset and tools generated in this study have the ability to offer a solid base for future insights into the underlying biology of conserved elements in the *An. gambiae* genome.

## Figures and Tables

**Figure 1 insects-12-00097-f001:**
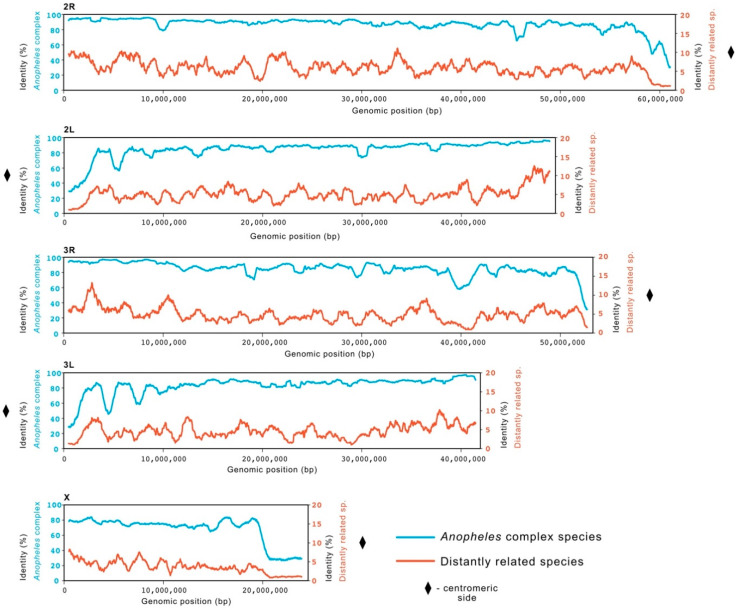
Conservation across *An. gambiae* chromosomes. A sliding window of length 1 Mb was used to calculate average identity (in percentage) for two groups of species in comparison to *An. gambiae*. A comparison with a group of five species of the *Anopheles gambiae* complex (in blue; left scale 0–100%), namely *An. coluzzii*, *An. arabiensis*, *An. quadriannulatus*, *An. melas* and *An. merus*, showed high average identity (81%) across the genome. The group of species more distantly related to *An. gambiae* (in red; right scale 0–20%), namely *An. darlingi*, *An. albimanus*, *Aedes aegypti*, *Culex quinquefasciatus*, *Drosophila melanogaster*, showed lower average identity (<5%). A decrease in conservation in both groups can be observed closer to the centromeric regions of each chromosome arm (represented by ♦).

**Figure 2 insects-12-00097-f002:**
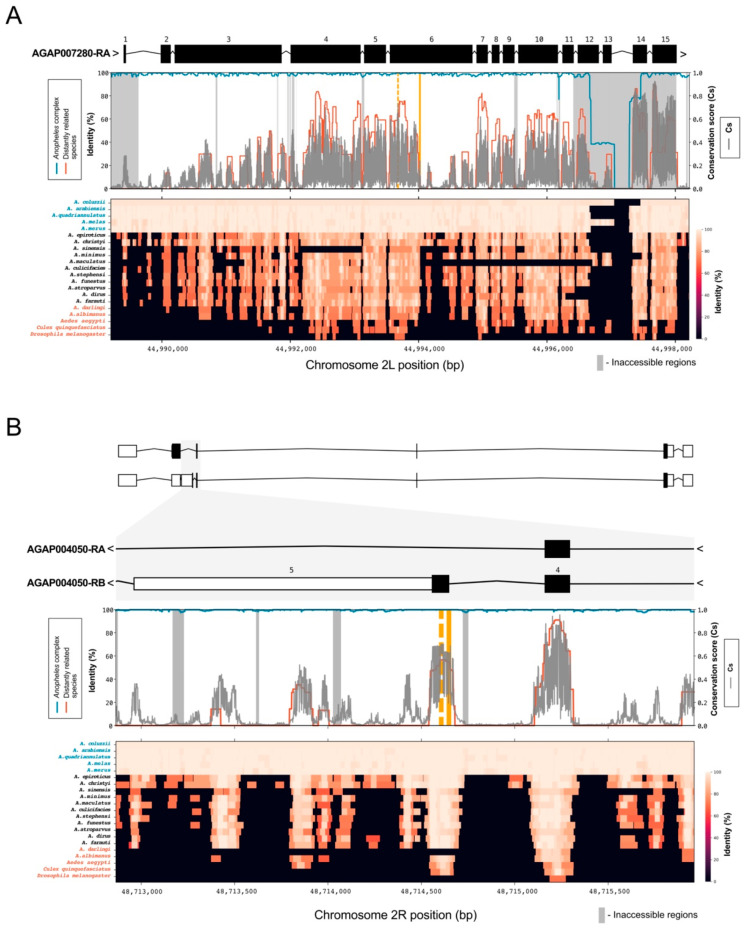
Conservation at *AGAP007280* and *AGAP004050* (*dsx*) gene drive target sites. (**A**) Top panel: the gene drive target site [9] in exon 6 of *AGAP007280* gene is highlighted (solid orange line). Middle panel: average identity (in percentage) was calculated for the five genome species within the *Anopheles gambiae* complex (blue line) and the genomes of the five most distantly related species included in the analysis (red line). Conservation score (Cs) in gray was also calculated, considering all 21 analyzed species and *An. gambiae* variation. The gray blocks are masked regions considered as inaccessible [22,23] due to the low confidence of genotyping. Bottom panel: heatmap represents the identity (in percentage) for each analyzed species. Species from the *An**. gambiae* complex are highlighted in blue, mosquito species most distantly related from *An. gambiae* are highlighted in red. (**B**) Target site for *AGAP004050* (*dsx*) falls between intron 4 and exon 5 [11] and it is marked in orange. With the exception of two species (*An. darlingi* and *D. melanogaster*), the target site shows high conservation (>70%) for the *Anopheles gambiae* complex and more distantly related species. The alternative target sites, identified in this study, are marked with a dashed orange line and show a higher conservation score.

**Table 1 insects-12-00097-t001:** Alternative gene drive target sites in the coding sequence of exon 5 in *dsx* gene. The existing gene drive target site is highlighted.

Chromosome	Start	End	Strand	Cs	Cs_max_	Cs_min_
2R	48714594	48714616	−	0.60	0.70	0.26
2R	48714589	48714611	−	0.56	0.70	0.20
2R	48714550	48714572	+	0.51	0.68	0.19
2R	48714551	48714573	+	0.50	0.68	0.19
2R	48714554	48714576	+	0.48	0.68	0.19
2R	48714561	48714583	−	0.45	0.68	0.15
2R	48714637	48714659	−	0.44	0.68	0.17
2R	48714640	48714662	−	0.41	0.68	0.17
2R	48714648	48714670	−	0.36	0.68	0.17

## Data Availability

The data presented in this study are openly available in Zenodo at 10.5281/zenodo.4304586, reference number 4304586.

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
