# Peer review of "Anopheles gambiae Genome Conservation as a Resource for Rational Gene Drive Target Site Selection"

_insects, 2021, doi:10.3390/insects12020097_

Round 1

Reviewer 1 Report

To aid the design of guide sequences for future gene-drive efforts, Kranjc et al. have performed an analysis of available population genetics data for Anopheles gambiae / coluzzi and genome assemblies for a range of dipteran species. Combining this into a single ‘conservation score’ they have identified a range of potential sites and further validated the site currently being targeted by the Target Malaria program. 

While the analyses appear sound there is a lack of available source code with which to validate them. Although the authors have provided a software package to retrieve results, they have not allowed a reviewer or other researchers to view the specific code. For a paper of this type, which does not generate any primary data, this should be provided to the reader.

The data chosen is appropriate with one small, but important, proviso; the AgamP3 (PEST) assembly is known to be a hybrid of Agam/Acoll. This has the propensity to miss variation specific to Agam in those regions where PEST is “coluzzi-like”. The S-Pimperena reference is poorly assembled, but should nevertheless be included in the analysis alongside PEST.  

The authors have included a broad range of dipterans - presumably arguing that conservation across the order means high conservation within the genus; however this might mean that regions highly conserved within anophelines but divergent from culicides would not be selected. This may be appropriate, but is not well argued within this paper. Examining the correlation of the conservation values given in Fig 1, and/or comparing the performance of the Cs statistic with and without the less-related dipterans would give more confidence that the inclusion of these species was appropriate. 

On a similar note the authors have used a population genetic dataset and a comparative genomic dataset to derive their statistic. Other species being targeted for gene drive (e.g. An. funestus, Ae aegypti) may only have one or the other of these. The authors have an opportunity to examine how well conservation between species relates to conservation within species. This would be an important improvement for people working on similar efforts in other species and should be included in this paper. 

Overall the paper is well written, the analyses are appropriate and the data chosen to do this is well suited to the task. I believe this would be a valuable addition to the field and of use to other researchers developing gene drives and I would recommend its publication in Insects. 

Minor comments below:

L147-149 - the assemblies used are in various states of completeness; how was missing data handled, and was there a floor level set for the number of assembled homologues needed to derive the statistic? As unassembled sequence might be particularly divergent, are the authors confident that inaccessibilty in Ag1000g correlates to inaccessibility in the wider spp?

L193 - it is unclear to me why normalisation was undertaken per chromosome arm, rather than across all autosomes or the whole genome. 

L246 - both X and Y axes should be properly labelled and arms shown in order (i.e. R/P before L/Q). As stated above a (QQ-like) comparison would be useful to see the appropriateness of combining conservation scores from these two groups

Reviewer 2 Report

The paper by Kranjc et al. describes a novel bioinformatic pipeline aimed at identifying highly conserved genomic regions in An. gambiae with potential functional or structural constraints, providing a quantitative parameter of conservation. The identification of functional conserved regions is key for employing effective CRISPR-Cas9 based techniques to locate and identify target sites (one or multiple) that affect gene expression and phenotypes, minimizing the potential of resistance.

The dataset and the bioinformatic tool could be of interest to those involved in the development of gene drive lines with the ultimate aim of vector control strategies. Overall the work is solid and straightforward; the conclusions are in accord with the data and writing is clear and succinct.

Minor:

  • revise italics and species names throughout the Ms and figures captions
  • I suggest to include a comparative description or literature data/available packages for other organisms (i.e. for model organisms) in the discussion section in order to sustain why the generated pipeline is novel, easy-to-use and accessible with respect to others already available. 
  • the layout and the style of the References list need substantial revision
  • please provide images with improved quality (Fig.1-2)
